# Sustainable Co-Creation Behavior in a Virtual Community: Antecedents and Moderating Effect of Participant's Perception of Own Expertise

**Natalia Rubio** [ID]**, Nieves Villaseñor * and MªJesús Yagüe** [ID]

Marketing Department, Universidad Autónoma of Madrid, 28049 Madrid, Spain; natalia.rubio@uam.es (N.R.); maria.yague@uam.es (M.Y.)

*   Correspondence: nieves.villasenor@uam.es

**Abstract:** Value co-creation by users in a virtual community is a key element of the community's value and sustainability. This paper first analyzes the effect on co-creation behavior of (1) users' altruism and (2) users' interactivity with the different firms housed in the virtual community. It considers different sustainable co-creation behaviors based on intensity, distinguishing between moderate or high intensity, where recommendation of the service represents a moderate level of co-creation and co-innovation at a high level. Both behaviors are oriented not only to the firms housed in the virtual community, but also to the virtual platform itself. Second, the study proposes whether users' perception of their expertise about the services and/or firms housed in the community exerts a moderating effect on the proposed modelling. Empirical contrast is performed using the virtual community TripAdvisor. This study contributes to current academic research on co-creation and sustainability. The results show that the user's altruism is the determining factor in encouraging moderate sustainable co-creation behavior (recommendation of the service), whereas user–firm interactivity is the critical antecedent for fostering high-level co-creation in the form of co-innovation activities. The results vary, however, depending on the segment of virtual community users analyzed (expert vs. non-expert). This study also contributes to formulation of business strategies to foster sustainable co-creation behavior with greater repercussions for long-term participation in the virtual community.

**Keywords:** sustainable co-creation; co-innovation; virtual community; interactivity; altruism

## 1. Introduction

The foundational premises of Service-Dominant Logic (SDL) provide a suitable framework for understanding how value is created on a digital platform, and more specifically in a virtual community [1]. Within the framework of this new logic [2,3], organizations do not create and deliver value to passive consumers; rather, value is co-created, or "jointly created," by multiple actors, such as stakeholders, firms, and customers [4], always including some beneficiary (axioms 6 and 10, respectively) [3].

A long line of research performed within the SDL framework focuses on analyzing the co-creation process in digital environments, identifying both antecedents and results of co-creation. Among antecedents, studies have considered both relationships with the medium in which the co-creation process develops (e.g., characteristics of the technology) and antecedents related to the actors involved, stressing the motivation that drives actors to participate in the co-creation process [5]. Among the consequences of the co-creation process, studies have considered primarily relational terms, such as satisfaction [6], loyalty, and brand equity [7–10].

Our literature review shows that none of the multiple studies on the process of co-creation in a virtual community models which antecedents of co-creation give rise to sustainable co-creation

behaviors that ensure long-term participation in the virtual community. This is not a trivial issue, as we are in a new era. Just as consumers increasingly stress questions such as wellbeing, social justice, and environmental factors like pollution or climate change in making purchase decisions, so also people intervene in the co-creation process because they are deeply motivated by the benefit they can contribute to other participants in the virtual community. Users in the virtual community participate in co-creation processes to be useful to and/or to provide a service and contribute to the wellbeing of other actors in the virtual community and to its sustainability. Although some studies in this direction [11,12] conclude that co-creation contributes to the sustainability of the ecosystem in which it takes place, greater understanding is needed of how the relationship between these variables occurs. This study's main theoretical contribution is thus to propose an integrative model of the value co-creation process and the antecedent variables that contribute to generating sustainable co-creation behavior. The study's second theoretical contribution proposes two levels of sustainable co-creation behavior: (1) A moderate level of co-creation represented by recommendation of the service to other users, and (2) a high level of co-creation reflected in the user's participation in co-innovation activities through the virtual community. A third theoretical contribution proposes the moderating effect of the user's perceived level of expertise on the relationship model proposed. It is currently a widespread management practice to recognize the expertise users share through virtual communities. For example, TripAdvisor awards badges that reflect its participants' knowledge level. Depending on the number and type of opinions shared, participants can be ranked as Senior Reviewers or experts in different services, be granted Passport Badges, Explorer Badges, etc.

This study's empirical contribution is its application of the theoretical model to virtual communities for services managed by third parties (hereafter, simply "virtual communities"). Although these platforms are the ideal service ecosystem for co-creation processes due to the interaction of multiple actors (e.g., firms housed in the platform, customers, potential customers, and virtual platform managers), research in this field has not analyzed how to encourage sustainable co-creation behavior on such platforms. Yet these platforms are currently gaining great importance due to advances of the internet and the greater accessibility enabled by smartphones, which permit new forms of interaction, communication, and consumption among individuals. Although customers' information at the time of deciding to make a purchase was historically limited to brand advertising, word of mouth, and rankings, purchasing choices are currently determined primarily by evaluations that appear in virtual communities. According to one study of [13], 91% of survey respondents were influenced by online opinions when choosing a restaurant or tourist lodging, and 94% stated that the opinions were sources of inspiration for them.

In sum, to date, the relationship between co-creation and sustainability has been studied from the perspective of strategic management [11,14] and applied to the areas of education [15], health [16], and tourism [17]. We find no studies, however, that specify how sustainable co-creation behavior is generated in virtual communities. The specific questions that guide this study are therefore: What factors motivate users of the virtual community to perform sustainable co-creation activities? What sustainable co-creation activities take place in a virtual community? (3) Since knowledge exchange is a fundamental asset of virtual platforms, does the user's perceived expertise have a moderating effect on the relationships proposed among the antecedents and sustainable co-creation behaviors?

Answering the questions proposed is very important for management. It will help managers to identify and focus their marketing efforts on the drivers that stimulate sustainable co-creation behavior. Ensuring survival of the virtual community requires knowing what motivates its members to participate in co-creation activities and what keeps them actively involved in the long term. The proposed modelling also enables us to orient recommendations for management to different recipients: Firms housed in the virtual community and managers of the virtual platform. Because results depend greatly on an effective value co-creation process, guidelines to foster sustainable co-creation behavior will improve virtual community users' satisfaction levels and increase the community's value, as well as the results of firms housed in the community.

The paper is structured as follows. The next section presents a summary of the relevant literature on sustainable value co-creation behaviors and factors impacting these co-creation behaviors. Next, we propose the research hypotheses. We then present the data and research methods, followed by a discussion of the main findings and managerial recommendations.

## 2. Theoretical Background

### 2.1. Understanding Sustainable Value Co-Creation Behavior in Virtual Communities

Early definitions of co-creation draw attention to customer participation in producing and delivering the product. We revise this conceptualization, claiming that co-creation goes beyond mere participation in collaborative work [18]. Various studies indicate that value co-creation goes beyond gathering the consumer's participation at any phase of the production process (co-production) and incorporates the actors' interactions, experiences, and social relationships that intervene in the process [19]. Ref [20] propose one of the models for the analysis of co-creation behaviors more often used by the scholars. Their co-creation scale comprises two dimensions: Customer participation behavior and customer citizenship behavior, with each dimension having four components. The elements of customer participation behavior include information seeking, information sharing, responsible behavior, and personal interaction, whereas the aspects of customer citizenship behavior are feedback, advocacy, helping, and tolerance. Virtual communities are ideal environments in which to manifest these new ways of participating in the co-creation process [1]. Although it is true that different co-creation behaviors exist for specific virtual environments (e.g., creating groups and/or events, participating in these groups or events, and sending or responding to invitations to "friend," these behaviors are typical of social networks [21–23]. The fundamental co-creation behavior in any virtual community, which is profoundly tied to the concept of value co-creation, involves users' participation, for example, generating and sharing content through recommendations and reviews [24–26]. More specifically, when a specific service is recommended in a virtual community, information and resources are transferred to the other users on the platform, encouraging the platform's functioning, as the recommendation increases the knowledge of those who view it. If the case involves potential consumers, these individuals benefit from the experience of the person who made the recommendation, as they can make better decisions. If it involves firms, the recommendation helps firms to have better knowledge of their customers' satisfaction level [20]. All information accumulated through recommendations generates unique, holistic value, ensuring sustainable competitive advantage. Recommendation of the service is thus considered a sustainable co-creation behavior in the Service-Dominant Logic (SDL) paradigm [27].

Co-innovation is another co-creation behavior stressed by researchers in digital environments [28–30]. Co-innovation in the virtual community is defined as the collaborative and proactive behavior oriented towards conceiving fresh ideas for design and development of new products, processes, and services, or the improvement of existing ones [31]. Ideation and collaboration are the cornerstone of co-innovation in virtual communities [32]. In the case of ideation, virtual community users share resources operating in the form of new ideas. In the case of collaboration, users identify problems in the products/services and propose solutions to them based on their experience and knowledge. Co-innovation stands out as a sustainable co-creation behavior because organizations can transform these ideas into concrete specifications for the product/service to improve the value proposal quickly and efficiently. They can reconvert a service or redefine a supply to adapt to users' changing tastes and new needs, ensuring better business results because of this process [33].

Most of the studies reviewed focus on co-innovation among consumers and firms [28,32,34], but the virtual platform itself is a necessary tool for the interaction among its participants and a crucial medium for co-innovation in the network era [31]. This study thus enriches prior studies by considering the importance of distinguishing between the co-innovation that occurs between users and service firms housed in the virtual community, and the co-innovation between users and managers

of the virtual community. This second type of co-innovation is defined by the same activities involved in user-service firm co-innovation, that is, the proposal of new ideas to improve the virtual platform (e.g., improving the interface, ease of use). Significantly, one of the most active forums on TripAdvisor is entitled "Help us make TripAdvisor better!" This forum allows TripAdvisor's users to propose ways to make participation in the virtual community more attractive. Finally, the combination of technology and knowledge shared by users in virtual communities encourages the co-creation process and thus the performance of the firms housed in the platform and of the platform itself. Through sustainable co-creation behavior (recommending the service and co-innovating oriented to the service firms and the platform), users pursue transparency and continuous improvement of the ecosystem that the virtual community represents.

## 2.2. Factors Impacting Co-Creation Behaviors

Several lines of research revolve around antecedents of co-creation behaviors. Some studies focus on traits of the virtual community that encourage these behaviors. These prior studies stress that aesthetic characteristics of the virtual community—for example, its capability to encourage interaction among participants and improve their feeling of social presence; or superior functional, utilitarian, or physical performance—give rise to greater perceived utility of the virtual community and thus foster value co-creation [35,36].

Studies of participants in the virtual community have distinguished a series of motivations—both extrinsic and intrinsic to individuals—as valid antecedents to co-creation behaviors [37]. Extrinsic motivations are primarily associated with external incentives for the participant. They include, but are not limited to, monetary rewards, and span a set of motivations such as reputation, recognition, status, and image projected to others. Intrinsic motivations are related to participants' needs and internal desires, and stress entertainment, pleasure, and altruism; intellectual stimulation; and a feeling of obligation to contribute. A third set of motivations includes extrinsic motivations that can be internalized when individuals transform external incentives into their own motives, for example, self-efficacy, learning, interactivity, and social motivations related to the search for emotional support, making friends, and strengthening relationships with the other participants in the virtual community. Multiple studies of antecedents of value co-creation include these motivations [34,38–42].

This study borrows two of the foregoing motivations, user altruism and user-service firm interactivity. In so doing, it follows attribution theory [43], an approach often used in studies on management and marketing. Attribution theory argues that favorable attitudes toward co-creation depend on how participants make attributions about the motives behind a co-creation. Overall, two types of motive attribution exist in the literature: Altruistic motives for virtual community wellbeing, and egoistical motives, or motives of self-interest. The motive of altruism reflects the user's desire to contribute to virtual community development. Egoism relates the community participant's ultimate goal directly to the participant's own interests, that is, to obtaining a direct response to his/her demands, comments, etc. from organizations housed in the virtual platform. This theoretical framework is appropriate for choosing the antecedents of co-creation behavior. Some authors, such as [44], stress the need to incorporate altruism as a necessary motivation for value co-creation. That is, they understand sustainable co-creation behavior in the virtual community as the result of the responsibility the participant acquires to the other stakeholders in the community. This responsibility is based on preventing damage to the community and acting proactively for the community's benefit [45]. As a theoretical reference for the proposed model, on the other hand, SDL affirms that the active dialogue that must be established in a co-creation process is necessarily interactive [3], and interactivity is a necessary characteristic of virtual communities [31,46].

## 2.3. Proposal of Hypotheses

Altruism has been considered as an important variable in encouraging individuals' willingness to share their knowledge in virtual environments [47]. Adapting the definition by [45], altruism is

the moral principle and standard that guides the user's participation in the virtual community. Altruistic users believe they have the responsibility to act appropriately in their relationships with the other actors in the virtual community [36,47], whether these are firms or other consumers. Users therefore seek to have their proactive participation in the virtual community contribute to its social wellbeing, making the community a place for fair and honest interaction. Altruistic participants pursue this goal above their personal interests. Altruism occurs when people enjoy helping others without obtaining anything in exchange. Users in the virtual community who care about altruistic issues provide value to other participants in the community—for example, potential buyers—by recommending products/services with which they have had good experiences and preventing those buyers from acquiring other products/services with worse performance [48]. Similarly, they wish both to help the firms housed in the virtual community improve their profitability and to contribute to improving the functioning of the virtual community itself [49,50]. The positive influence of user's altruism on participation in co-innovation activities has been contrasted in studies of open source communities [51,52]. The altruistic user contributes ideas in the form of new concepts of the product and/or service, new characteristics of services, or new service categories, motivated by contributing to better functioning of the virtual community and of the actors who operate in it [34]. Based on the foregoing, we propose:

**Hypothesis 1a (H1a).** *The user's altruism positively influences recommendation of the service to other users in the virtual community.*

**Hypothesis 1b (H1b).** *The user's altruism positively influences co-innovation with service firms in the virtual community.*

**Hypothesis 1c (H1c).** *The user's altruism positively influences co-innovation addressed to managers in the virtual community.*

According to [53], the second antecedent proposed, interactivity, can be tackled from three different perspectives: (1) As a feature of technology, or the medium that provides the human-to-human and human-to-computer communication; (2) as a user's perception after using a technology or going through a process (thus, a subjective evaluation by the user rather than an evaluation based on the objective features of the website; use of this focus has been based on the argument that measurement of interactivity is more closely related to how users perceive and/or experience technological features than to provision of these features; and finally, (3) as a process of message exchange, in which interactivity is defined as two-way communication between sender and receiver [54]. This study adopts the third definition, since SDL argues that interactivity plays a fundamental role in the success of most collaboration activities [55]. One manifestation of two-way communication is the response of firms housed in the virtual community to comments, suggestions, complaints, etc. by the other participants on the platform [1].

As mentioned above, effective communication between firms and the other users of the virtual community assumes a significant drive to involve other users and thus to increase opportunities to co-create value, both conceptually [56,57] and empirically [58–61]. Social exchange theory grounds the next set of hypotheses. This theory argues that individuals invest more effort in a social exchange activity—in this case, sustainable co-creation behavior—if they perceive greater response from the firms [62].

First, user-service firm interactivity in the virtual community encourages users' comprehension and understanding of the organization. Such understanding can then be transformed into a favorable opinion of the products/services commercialized and better organizational reputation [60]. More frequent, faster, richer communication between user and company in the virtual community are passionately and emotionally linked to both participants [63], enhance the feeling of being engaged with the organization [61], and drive users to participate in the value co-creation process, supporting the firm by recommending its brands, products, and/or services.

Interactivity is also manifested in the responses firms give to the requests of other virtual community users, who enter reciprocally into dialogue that encourages value co-creation by sharing relevant information with the firms [64]. User-service firm interactivity enables the virtual community to generate ideas about new products and services [65], and encourages user feedback to organizations about mistakes, areas for improvement, and possible solutions for the products/services they provide [42,66]. Further, interactivity, measured through organizations' response to users, fosters user response. Interested in gaining more control of the interaction and in making communication through the virtual platform more dynamic, users are inclined to participate in co-innovation activities for the virtual platform itself [66–69]. For example, one can see from the TripAdvisor forum that the possibility of encouraging user-service firm interactivity has generated comments from participants who seek to adapt the virtual platform more to their communication needs. For example, users propose that TripAdvisor managers be given the right to reply when the organization responds to user comments, or request incorporation of new languages, new criteria for viewing their comments, etc. Based on these arguments, we propose the following:

**Hypothesis 2a (H2a).** *User-service firm interactivity positively influences recommendation of the service to other users in the virtual community.*

**Hypothesis 2b (H2b).** *User-service firm interactivity positively influences co-innovation with service firms in the virtual community.*

**Hypothesis 2c (H2c).** *User-service firm interactivity positively influences co-innovation addressed to managers in the virtual community.*

*2.4. Moderating Effect of Perceived Expertise Level of the Virtual Community User*

Finally, in contexts mediated by technology, as in the case of virtual communities, numerous studies assert that the participant's expertise is a critical resource that encourages the process of value co-creation [36,70,71]. Expertise has been considered in general primarily in relation to internet navigation [72], and specifically in relation to a specific technology [36], as well as to the product/service [71]. This study analyzes the perceived level of expertise about a product/service, defined as "the ability to perform product or service related tasks successfully" [71]. In keeping with Bandura's social-cognitive theory (SCT) (1977), individuals identify their operating resources, including their level of expertise, and this identification conditions their decisions about what activities to perform, what level of effort and persistence they must assume to perform these activities, and what results they expect from them. Positive perception of their expertise level–which is positively connected to their self-efficacy, in overall terms–reflects individuals' perception of their capacity to organize and implement specific actions that lead to certain levels of results [73]. This perception is important in virtual communities oriented to sharing knowledge. SCT argues that having little confidence in one's level of expertise constitutes a barrier to participants' sharing their knowledge with the other members of the community. Individuals with more confidence in their expertise will, in contrast, participate in co-creation activities [74]. Studies of value co-creation in digital environments have thus investigated primarily the direct effect of expertise on knowledge sharing [36].

In contrast to prior research, this study tackles the moderating effect of perceived level of expertise on the causal relationships proposed. According to SCT theory, participants' desire to contribute and add value to the community must be strengthened by their capability to achieve this value [36]. That is, perceived level of expertise is a very important intrinsic motivating factor in the co-creation process [75], as it strengthens the altruism participants express in the virtual community. Specifically, insofar as users of the virtual community care about other participants on the platform (e.g., want to help others make better decisions or help firms improve their results), they recognize the competence they derive from their expertise and feel more secure than do participants with less expertise about

the information and opinions they share [76]. This feeling makes participants more likely both to make critical comments [42,77] and to make recommendations about a service and/or brand [70,78]. Similarly, [79] argue that people with greater technical knowledge of a product/service—for example, those who use it more frequently—are more likely not only to adopt innovations, but to generate them [80]. Prior studies have demonstrated that users with greater expertise about a product/service show more skill in contributing to its improvement [81], since they detect problems and suggest solutions. These individuals also show greater capability to contribute original and innovative ideas [80,82], and this greater level of self-perceived expertise strengthens the influence of altruism on participation in co-innovation activities in the virtual community. Further, when altruistic people believe they have more expertise, they are more likely to perform behavior that is challenging for them (e.g., an intellectual challenge such as proposing goals that are difficult to achieve). People with more expertise more frequently respond to organizations by providing creative and/or original ideas, creating trends, etc. In sum, we propose level of expertise as a characteristic of virtual community participants that can strengthen their altruism, encouraging co-creation behavior. Based on the foregoing, we propose:

**Hypothesis 3a (H3a).** *The user's perceived level of his/her expertise has a positive moderating effect on the relationships between altruism and sustainable co-creation behaviors in the virtual community.*

This study also seeks to analyze whether perceived expertise exerts an additional moderating effect in the influence of user-service firm interactivity on sustainable co-creation behavior. Prior studies have identified a positive moderating effect of expertise on the relationship established between interactivity in the virtual community and variables of results, such as trust, popularity, and reputation of organizations that operate in the virtual community [74].

According to [83], virtual community users must generally perceive reciprocity to perform co-creation behavior, but such interactivity is an especially important antecedent of co-creation for users who evaluate their level of expertise positively. First, users with greater perception of their expertise feel more autonomy and authority to make their interactivity with firms in the virtual community a priority than do users with less expertise. Organizations that respond to these more expert users' demands receive as a reward the positive evaluation and recommendation of their products/services such users make. As these users also reveal their level of expertise to the other users in the virtual community, being recognized as a valuable source of information is an additional motivation for them to participate in value co-creation processes [84]. Second, users with a higher level of expertise show more egocentric behavior. Since they are more aware of the quality of the information they share and of their capability to contribute solutions and novel ideas and improve products and/or services, they demand more interaction from the organizations housed in the virtual community as a requirement for making the effort involved in high-level co-creation behavior–that is, participation in co-innovation activities. Based on the foregoing, we propose:

**Hypothesis 3b (H3b).** *The level of user's perceived expertise has a positive moderating effect on the relationship between user-service firm interactivity and sustainable co-creation behavior in the virtual community.*

Next, the theoretical framework shown in Figure 1 is proposed.

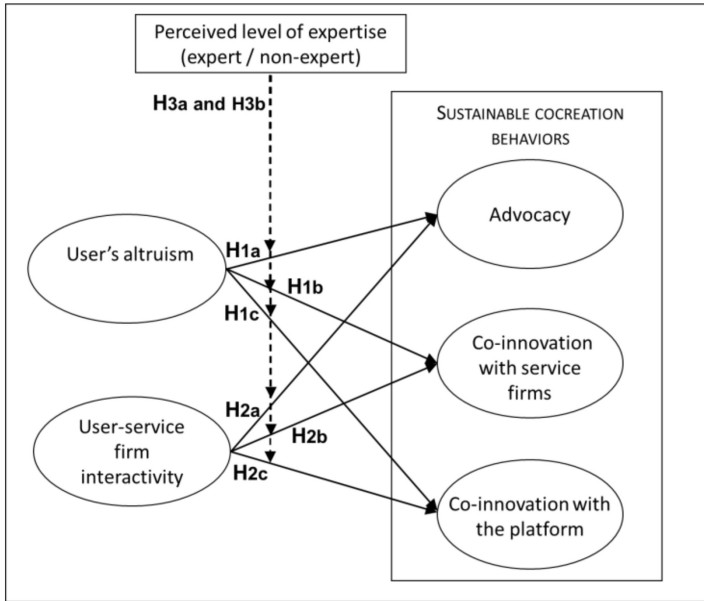

**Figure 1.** Theoretical research model.

## 3. Methodology

To achieve the research goals proposed, we perform an empirical study, taking a sample of 600 TripAdvisor users who stated that they shared their service experiences through TripAdvisor's virtual community. The information was obtained by telephone questionnaire.

The sample profile shows that 45.2% of the survey respondents comment 50% or more of the times they use the platform. As to type of service on which they comment most frequently, 74.2% of the sample comment on hotels, followed by 49.5% who comment on restaurants. Others use the virtual community to comment on flights and vacation rentals.

The Table 1 shows the sample profile for gender, age, education level, family income, and type of family.

**Table 1.** Sample profile.

| Age | Gender | Education Level | Family Income | Family Size |
|---|---|---|---|---|
| <25 years old: 17.7% | Men: 49.7% | Primary school: 10.8% | At least 1000 euros/month: 11.2% | 2-member families without children: 41% |
| 26–35: 29% | Women: 50.3% | Secondary school: 39.3% | 1000–2000 euros: 51.8% | Families with one child: 23.3% |
| 36–50: 37.2% | - | University: 49.5% | 2001–3000 euros: 29.7% | Families with two children: 19.5% |
| >51: 16.2% | - | - | >3000 euros: 6.8% | - |

The items used to measure the concepts were obtained by adapting scales employed previously in the academic literature. First, of the antecedents proposed, altruism of the virtual community user was measured by adapting the scale in [47]. User-service firm interactivity was measured by adapting the scale in [85]. As to the results, sustainable co-creation behavior in the form of recommendation of the service was measured by adapting the items used in [20]. Co-innovation with service firms and with the virtual platform was measured by adapting the scale developed by [25]. Finally, perceived level of expertise was measured through the three statements from the scale developed by [86]: "I consider myself an expert in use of the services TripAdvisor provides", "I consider myself an expert in the services on which I comment on TripAdvisor", and "I consider myself an expert in the firms that offer their services on Tripadvisor". We obtained an Alpha Cronbach of 0.73 for this scale, guaranteeing

its reliability. All scales were measured using 10-point Likert scales that ranged from 0 (disagree completely) to 10 (agree completely).

The first techniques used to analyze the information were descriptive analysis of the model variables. Next, we conducted covariance-based confirmatory factor analyses and structural equation modelling with AMOS 25 software. The analysis used the data aggregated for all virtual community users in the study (n = 600). Structural equations modelling methodology was chosen due to: (i) The complex relations among the different model variables, and (ii) the need to test the theory of the model [1]. We used AMOS software because it provides data about the fit of the global model (AMOS seeks to test a solid theory, which is the main objective of this work: Confirmatory analysis), as opposed to other software such as PLS, which is better suited for predictive and theory development applications (exploratory analysis) [87].

Third, to analyze the moderating effect of perceived level of expertise, multi-group analyses of the structural modelling of covariance were performed using the program AMOS 25. For the multi-group analyses, we classified the virtual community users into two segments, those who considered themselves experts in the firms and/or services housed in the virtual community (n = 303) and those who did not consider themselves experts on the topic (n = 297). We identified these two segments by applying two-step cluster analysis to the survey respondents' replies on the three items used to measure perceived level of expertise.

## 4. Results

### 4.1. Measurement Model

As recommended by [87], we confirmed the quality of the measurement scales for each sample. We performed confirmatory factor analysis using the AMOS 25.0 program. The results for both samples showed a very satisfactory fit for modeling the four factors proposed. In the sample of expert users, the relationship $\chi^2$/d.f. was 1.94, close to the maximum threshold of 2 recommended by [88]. The CFI and AGFI values were 0.98 and 0.90, respectively, higher than the minimum value of 0.9 recommended by [88]. The RMSEA was 0.05, lower than the maximum of 0.06 suggested by [88]. In the sample of non-expert users, the values obtained also respect the limits recommended in the academic literature ($\chi^2$/d.f. = 1.95; CFI = 0.98; AGFI = 0.89; RMSEA = 0.06). (CFI = Comparative fit index; AGFI = Adjusted goodness of fit index; RMSEA = Root mean square error of approximation; $\chi^2$ = Chi-squared; d.f. = degree of freedom; NFI = Normed fit index).

Table 2 show the results for reliability and validity in both samples (expert and non-expert users). In all cases, the statistics for reliability—Alpha Cronbach and composite reliability—were higher than the minimum value of 0.70 [89]. Variance extracted was greater than or equal to 0.5 in all cases, and all items had sufficient convergent validity, as all parameters were statistically significant. We also confirmed discriminant validity in both samples. Table 3 shows the root of the percentage of variance extracted for each construct, which in all cases is higher than the correlation between each pair of concepts.

Finally, we examined measurement invariance between the two groups, as follows. First, we performed multigroup confirmatory analysis and found that the results showed satisfactory fit ($\chi^2$ = 388.31; d.f. = 198; $\chi^2$/d.f. = 1.96; CFI = 0.98; NFI = 0.95; GFI = 0.93; AGFI = 0.89; RMSEA = 0.04). Second, we imposed the restriction of equality of parameters for the two samples and compared the goodness of fit results for the restricted and unrestricted models ($\Delta\chi^2$ = 13.07; $\Delta$d.f. = 8; $p$ = 0.11 > 0.01). Since the model's fit did not worsen significantly, this result guarantees measurement invariance. Thus, the differences observed between the causal relationship models will be due to the causal relationships themselves and not to measurement of the constructs.

**Table 2.** Analysis of reliability and validity of measurement scales for experts and non-experts.

| Variables | | $L_i$ ex | $L_i$ nex | $E_i$ ex | $E_i$ nex | Reliability — Cronbach's Alpha ex | Reliability — Cronbach's Alpha nex | Reliability — Composite Reliability (CR) ex | Reliability — Composite Reliability (CR) nex | Validity — Average Variance Extracted (AVE) ex | Validity — Average Variance Extracted (AVE) nex | Validity — Convergent Validity ex | Validity — Convergent Validity nex |
|---|---|---|---|---|---|---|---|---|---|---|---|---|---|
| **User's altruism: When I express an opinion on TripAdvisor about a service I have used,** | $v_1$: I give other users useful advice | 0.88 | 0.90 | 0.23 | 0.20 | | | | | | | t = 12.59 *** | t = 17.92 *** |
| | $v_2$: I teach other users to choose the service better | 0.75 | 0.89 | 0.44 | 0.20 | 0.84 | 0.83 | 0.91 | 0.85 | 0.79 | 0.67 | t = … | t = … |
| | $v_3$: I help other users | 0.88 | 0.63 | 0.23 | 0.60 | | | | | | | t = 12.46 *** | t = 10.50 *** |
| **User-service firm interactivity: The firms about which I express opinions on TripAdvisor** | $v_4$: Answer me when I communicate directly with them | 0.81 | 0.70 | 0.35 | 0.52 | | | | | | | t = 14.90 *** | t = 11.90 *** |
| | $v_5$: Respond to my comments about service | 0.79 | 0.82 | 0.37 | 0.32 | 0.85 | 0.82 | 0.85 | 0.83 | 0.65 | 0.63 | t = … | t = … |
| | $v_6$: If I encounter errors, I communicate them (e.g., defective packaging or merchandise in the wrong place, products past the use-by date on the shelves, etc.) | 0.82 | 0.85 | 0.33 | 0.28 | | | | | | | t = 15.38 *** | t = 16.22 *** |
| **Advocacy: When I express an opinion on TripAdvisor about a service I have used,** | $v_7$: If I liked it, I say positive things about it | 0.92 | 0.86 | 0.15 | 0.26 | | | | | | | t = … | t = … |
| | $v_8$: If I like it, I recommend it | 0.88 | 0.85 | 0.22 | 0.28 | 0.90 | 0.82 | 0.90 | 0.83 | 0.76 | 0.63 | t = 22.37 *** | t = 15.59 *** |
| | $v_9$: If I like it, I encourage others to use it | 0.81 | 0.65 | 0.35 | 0.58 | | | | | | | t = 18.28 *** | t = 11.05 *** |
| **Co-innovation with service firms: On TripAdvisor, I propose** | $v_{10}$: New modes of service | 0.93 | 0.90 | 0.13 | 0.19 | | | | | | | t = … | t = … |
| | $v_{11}$: Ways of improving existing services | 0.89 | 0.87 | 0.19 | 0.25 | 0.93 | 0.92 | 0.93 | 0.92 | 0.83 | 0.79 | t = 26.77 *** | t = 22.19 *** |
| | $v_{12}$: Ideas to identify trends | 0.89 | 0.89 | 0.20 | 0.20 | | | | | | | t = 25.67 *** | t = 23.26 *** |
| **Co-innovation with the platform: I collaborate with TripAdvisor on the following actions to improve:** | $v_{13}$: Its informative content | 0.87 | 0.92 | 0.25 | 0.15 | | | | | | | t = … | t = … |
| | $v_{14}$: Its aesthetic | 0.79 | 0.83 | 0.37 | 0.32 | 0.88 | 0.90 | 0.88 | 0.90 | 0.71 | 0.75 | t = 16.90 *** | t = 20.43 *** |
| | $v_{15}$: Its ease of use | 0.87 | 0.85 | 0.25 | 0.28 | | | | | | | t = 19.45 *** | t = 21.70 *** |

**Note**: EX = experts; NEX = non-experts; r = correlation between Li; $L_i$. = Standardized loading; $E_i = (1 - R^2)$: Error variance.

**Table 3.** Analysis of discriminant validity for both pairs of samples (users with expertise on the service vs. non-expert users) by average variance method.

| Variables | Segments | User's Altruism | User-Service Firm Interactivity | Advocacy | Co-Innovation with Service Firms | Co-Innovation with the Platform |
|---|---|---|---|---|---|---|
| User's altruism | Expert | **0.89** | 0.72 | 0.58 | 0.56 | 0.56 |
| | Non-expert | **0.82** | 0.70 | 0.57 | 0.61 | 0.55 |
| User-service firm interactivity | Expert | 0.67 | **0.81** | 0.38 | 0.67 | 0.69 |
| | Non-expert | 0.63 | **0.79** | 0.54 | 0.56 | 0.51 |
| Advocacy | Expert | 0.62 | 0.43 | **0.87** | 0.22 | 0.43 |
| | Non-expert | 0.55 | 0.55 | **0.79** | 0.27 | 0.19 |
| Co-innovation with service firms | Expert | 0.51 | 0.66 | 0.25 | **0.91** | 0.65 |
| | Non-expert | 0.60 | 0.56 | 0.30 | **0.89** | 0.78 |
| Co-innovation with the platform | Expert | 0.53 | 0.70 | 0.43 | 0.67 | **0.84** |
| | Non-expert | 0.53 | 0.52 | 0.23 | 0.77 | **0.87** |

**Note:** Values on the diagonal correspond to the square root of the average variance extracted in each construct. Values below the diagonal represent the correlations between pairs of constructs. Values above the diagonal represent HTMT values.

## 4.2. Causal Relationship Model

We estimated the model in Figure 1 using structural equations modeling, omitting the moderating effect. The fit obtained is satisfactory ($\chi^2$ = 108.95; d.f. = 47; $\chi^2$/d.f. = 1.88; CFI = 0.99; NFI = 0.98; IFI = 0.99; GFI = 0.98; AGFI = 0.95; RMSEA = 0.04), and all proposed hypotheses are confirmed. Figure 2 presents the path coefficients for the overall model. We see that the most significant antecedent for encouraging recommendation of the service is altruism, while user-service firm interactivity plays a more important role in fostering high levels of co-creation: Co-innovation with service firms and co-innovation with the virtual platform itself.

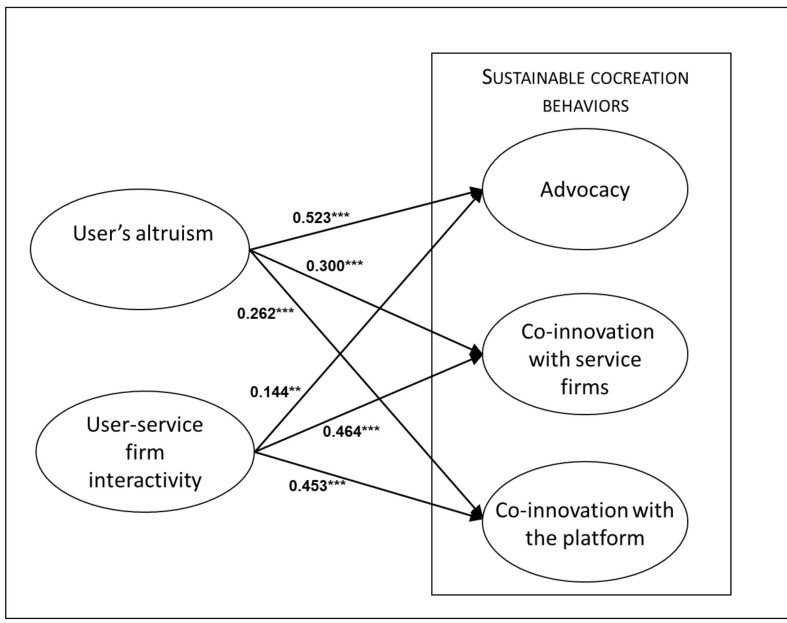

**Figure 2.** Estimation of Model Ratios. **Note:** *** $p < 0.01$; ** $p < 0.05$; * $p < 0.1$.

We now consider the moderating effect and perform multigroup structural analysis for expert and non-expert users. We compare the results of the two models—the first, unrestricted model, and a

second model, on which we impose the restriction of equality for the structural parameters in the two segments (restricted model). The goodness of fit results worsen significantly in the model when we impose restrictions of equality on the structural relationships, suggesting that some restrictions cannot be sustained and confirming that user's perceived level of expertise exerts a moderating effect on the relationships proposed in the model (see Table 4). That is, intensity of the causal relationships proposed in $H_1$ and $H_2$ is significantly different for expert vs. non-expert users.

**Table 4.** Multigroup structural analysis results. CFI = Comparative fit index; AGFI = Adjusted goodness of fit index; RMSEA = Root mean square error of approximation; χ2 = Chi-squared; d.f. = degree of freedom.

| | $\chi^2$ (d.f.) | $\chi^2$/d.f. | $\Delta\chi^2$ ($\Delta$df) | p | CFI | GFI | AGFI | RMSEA |
|---|---|---|---|---|---|---|---|---|
| **Structural model without restrictions** | 34.79 (9) | 1.90 | - | - | 0.98 | 0.95 | 0.91 | 0.03 |
| **Structural model with restricted parameters** | 52.39 (15) | 1.95 | 17.60(6) | 0.007 | 0.97 | 0.94 | 0.91 | 0.04 |

$H_{3a}$ is partially confirmed, since, as proposed, the effect of user's altruism on recommendation of the service is significantly stronger in the case of experts. Contrary to the hypothesis proposed, however, the sample of non-expert users shows significantly more intense influence of altruism on co-innovation with service firms and co-innovation with the platform.

We also partially confirm $H_{3b}$. Specifically, we find that the positive influence of user-service firm interactivity on co-innovation with service firms and co-innovation with the virtual platform is significantly stronger for expert users. Contrary to our hypothesis, however, the influence of user-service firm interactivity on recommendation of the service is significantly more intense in the segment of non-expert users (see Figure 3). These results are discussed in the next section.

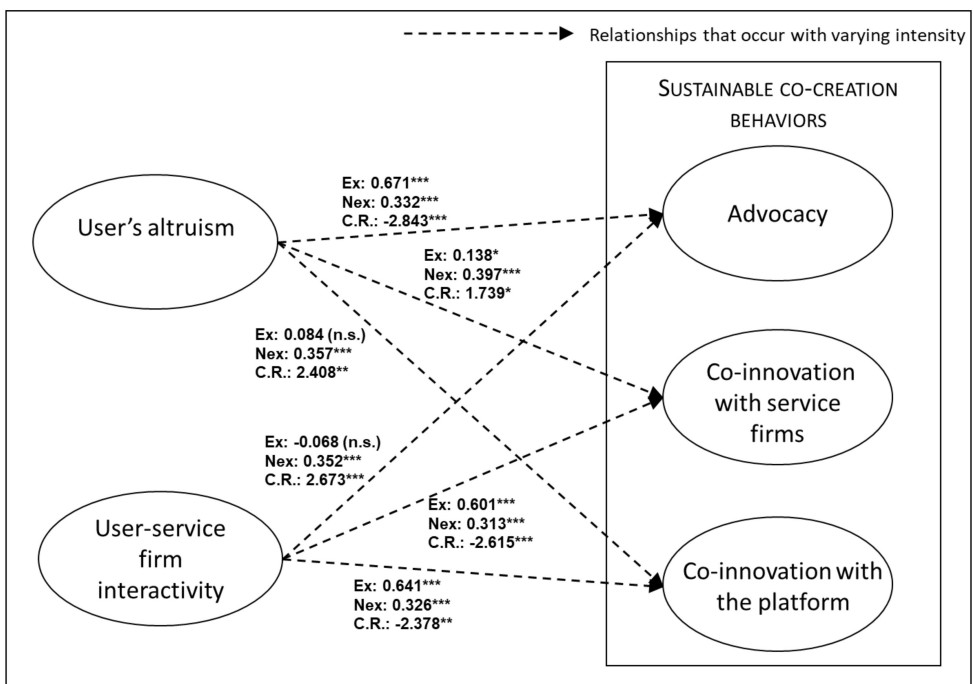

**Figure 3.** Results of multigroup analysis for experts and non-experts. **Note:** Ex = expert users; Nex = non-expert users. *** $p < 0.01$; ** $p < 0.05$; * $p < 0.1$; n.s. = not significant; C.R.: Critical ratio of differences between parameters. t = 1.65 for $p < 0.10$ *; 1.96 for $p < 0.05$ **, and t = 2.58 for $p < 0.01$ ***.

## 5. Discussion and Conclusions

The first theoretical implication of this study is the confirmation of a conceptual framework that supports the structural relationships established between the user's altruism and user-service firm interactivity in the virtual community regarding sustainable co-creation behavior. This study also enriches prior studies on the co-creation process in virtual communities by identifying two levels of co-creation behavior based on the effort required of the participant to perform it: Recommendation of the service, which represents a moderate level of co-creation, and participation in co-innovation activities oriented both to firms housed in the virtual community and to the virtual platform itself, which represents a high level of co-creation. Since prior studies of co-creation behavior in virtual communities have not analyzed these two behaviors together [21–24], our study represents an advance in this direction.

First, the results of the general model confirm that both user's altruism and user-service firm interactivity are appropriate antecedents for fostering sustainable co-creation behavior in the virtual community. This result is consistent with prior studies [38,39]. In exploring our results in greater depth, however, we see that altruism is the most significant antecedent for encouraging recommendation of the service, but that user-service firm interactivity is the most prominent antecedent for encouraging co-innovation with firms in the virtual community and with the virtual community itself. That is, participants in the co-creation process only undertake the greater cognitive effort involved in performing co-innovation activities if they receive a response from the organizations.

Second, this study represents a significant and original advance over prior research on the moderating effect of perceived level of expertise in the co-creation process [84], as prior studies do not propose either the antecedents analyzed here or the sustainable co-creation behaviors identified. Studying the variable expertise is of great interest to the virtual community because expertise shared by the community's participants is the main asset for this type of digital environment.

According to the results obtained in the multigroup analysis, virtual community users' level of expertise moderates the effect of the antecedents analyzed on sustainable co-creation behavior. Prior studies [90] already indicate that reasons for performing co-creation behavior may vary in strength depending on user's level of expertise. In the group of non-expert users, altruism and interactivity were equally important antecedents stimulating the sustainable co-creation behaviors proposed. In the case of expert users, however, altruism encouraged only the moderate level of co-creation (recommendation of the service) and had no effect on co-innovation. Expert users must perceive interactivity with service firms housed in the virtual community to participate in co-innovation activities. These results agree with those obtained by [63], according to which virtual community users with a higher level of expertise discriminate in the audience they address when they participate in the co-creation process. Their altruism moves them to perform moderate co-creation behavior: Recommendation of the service. Because expert users believe that other users (primarily customers or potential customers) have a lower level of expertise, they think they should help other users by giving recommendations. Leader–member exchange theory [91] argues, however, that users with greater expertise will only invest the greater effort (both cognitive and in terms of time) required to participate in high-level co-creation through co-innovation activities if they perceive interactivity with the firm, a response these users consider as better in terms of their power (e.g., power of reward). Expert users participate in co-innovation activities primarily motivated by the enriching interactivity firms provide. If they believe this interactivity will result in mutual benefit in a quality relationship, expert users will undertake these co-innovation activities.

Comparison of the two segments (experts vs. non-experts in services/firms), in the hypotheses proposed (H3a and H3b) indicates that the positive effect of altruism and interactivity on sustainable co-creation behavior must be stronger among expert users, based on the fact that this group's security in its expertise is an additional motivation strengthening the relationships proposed. The results of these hypotheses are only partially fulfilled, however, since the effect of the user's altruism on recommendation of the service is significantly stronger in the case of experts. The effect of interactivity

on both levels of co-innovation proposed is also stronger in this group. Contrary to our expectations, however, altruism influences both types of co-innovation with significantly more intensity in the sample of non-expert users than in the sample of experts. This finding is consistent with the research of [84], who explain that people who view themselves as less expert are generous with the resources they share in virtual environments—in this case, ideas for co-innovation with service firms and with the platform. They do not require anything in exchange because they do not believe that they have made an important contribution. As users' feeling of expertise increases, however, they believe that their contribution is extremely valuable to co-innovation. They feel highly qualified to co-innovate and demand rewards for their participation in the co-creation process (e.g., recognition by the firms). Again, contrary to our proposal in H$_{3b}$, the effect of interactivity on recommendation of the service is significantly stronger in the case of non-experts. This result confirms the theory of [92], who argue that less-expert users of the virtual community rely more than non-experts on interaction with the organizations as a non-functional attribute to stimulate recommendation of their products/services. Non-expert users show higher levels of appreciation for the organizations with which they interact and from which they obtain an answer to their questions, requests, etc. When they receive this response, therefore, they recommend the service out of gratitude and trust, as they do not feel they have sufficient ability to recommend the service based on more technical evaluations of the products/services. Users with greater expertise believe, however, that they have sufficient ability to evaluate the objective characteristics of the firms' products/services and make their service recommendations based on these characteristics, without depending on interactivity with the firms.

According to the results of the study, it should be noted that it is relevant to increase investments efforts in fostering processes of value co-creation with virtual community participants. In particular, the importance of altruism must be enhanced in order to contribute to the proper functioning of the virtual community. Additionally, given the key role of the user-service firm interactivity to develop sustainable co-creation behaviors, maximal effort should also be devoted to ensure that the service firms in the virtual community communicate with and listen to virtual community participants, interact with them and implement an active dialogue to share and exchange ideas with them. Advocacy and co-innovation with service firms and with the platform are sustainable co-creation behaviors that positively affect the virtual community user's perceived value and virtual community's brand equity.

Our results provide useful insights for the various users of a virtual community. Users' active participation is fundamental to ensuring the community's sustainability and has been identified as an important measure of online community performance. This study thus directly targets the key factors that strengthen this participation by incorporating recommendations for both service firms housed in the virtual community and managers of the virtual community. First, the results show that interactivity is fundamental to encouraging co-innovation with service firms and with the virtual platform itself. We have seen that firms in the virtual community usually have reactive behavior and respond to users who make recommendations or positive comments by thanking them and attempting to justify the firm when they receive criticisms in the virtual community. This study proposes, however, that firms in the virtual community should be proactive in seeking interaction with their users. Firms can do this by using their comments in the virtual community to motivate users who have tried their services to propose ideas on how to improve the service, what new services would be interesting to incorporate, etc. Further, firms must take advantage of customers' visits to their establishments (hotels, restaurants, etc.) to encourage customers to communicate their experience to the virtual community, not only as a recommendation, but also through participation in co-innovation activities. To do so, firms can organize a lottery, awarding a prize (e.g., discount on next visit, gifts) to the best ideas gathered through the virtual community about a new service, improvement of service, etc. Such action enables the firms housed in the virtual community to learn about market trends and incorporate them into their service offerings to provide greater value to their customers, simultaneously improving the value of their business and their profitability.

Managers of the virtual community can also contribute to improving user-service firm interaction. For example, we propose introducing new functions on the virtual platform. Just as virtual community users can indicate which comments seem most useful to them, firms can develop a metric by which they vote through the virtual community on which comments, ideas, etc. from users who are customers or potential customers seem useful to them. Such a metric is especially interesting for involving expert users in co-innovation activities, since interactivity with firms is vital to these users. This measure would also stimulate less-expert users to be more inclined to recommend the firms' services. Further, managers of the virtual community could implement tools to make user-service firm interaction more dynamic. Here, we propose that the virtual community allow direct comments among users, and between users and firms to make communication faster and more efficient and to increase the feeling of interactivity, as occurs in other digital environments, such as Facebook, Twitter, and Instagram. Most virtual communities currently allow people to send messages that other members of the community view or to ask firms questions in specific sections. In light of this study's results, however, and given the importance of interactivity in fostering recommendation and co-innovation, we believe it is beneficial for the virtual community to incorporate direct communication tools through comments among different types of user (an action more characteristic of social networks to date).

On the other hand, the results show that the user's altruism is relevant for encouraging experts to recommend the service and non-experts to recommend and co-innovate. Although altruism is an intrinsic motivation of the user, managers of the virtual community can perform some actions that strengthen altruistic feeling and promote co-creation behavior. For example, they can attempt to encourage users not only to indicate whether or not a comment was useful for them, but also to detail through direct comments what was most useful among the comments they viewed. When expert users see the direct mentions their comments have generated, they will feel they have helped other users make better decisions and will continue to participate in the co-creation process. Less-expert users, also motivated by altruism but not secure about their comments, will have their confidence strengthened to keep participating in co-creation activities as their comments generate direct mentions that highlight the benefits they have contributed to the community through their co-creation.

In sum, the foregoing paragraphs provide guidelines that both firms and managers of the virtual community can implement to strengthen users' altruism and user-service firm interactivity to keep members of the virtual community involved in the long term through sustainable co-creation behaviors, such as recommendation of the service and co-innovation with service firms and with the platform.

Finally, we must note some limitations of this study, which can give rise to future lines of research. First, as the co-creation behaviors were obtained through answers by survey respondents based on their perceptions, it is advisable to follow up by analyzing respondents' real behavior in the virtual community. Next, as the study was performed with a group of participants representative of the Spanish market, repetition in other geographic contexts would give it greater external validity. Future studies could also consider including new antecedents in the modeling proposed (e.g., other motivations of virtual community users, such as hedonic benefit). They could also propose which effects sustainable co-creation behaviors (recommendation of the service and co-innovation) have on the variables of results related to the virtual community (e.g., user's identification with the virtual community, sustainability, brand equity of the virtual community, etc.). Finally, it would be interesting to study the moderating effect of specific variables, such as age, sex, income, and education on relationships among the antecedents analyzed here and sustainable co-creation behaviors, as well as to contrast whether the proposed modelling can be applied to other types of virtual community.

**Author Contributions:** Conceptualization, visualization, writing—original draft, writing—review & editing, N.V. and N.R.; conceptualization, project administration, supervision, visualization, writing—review & editing, M.Y. and N.R.; project administration, supervision, visualization, writing—review & editing, M.Y. All authors have read and agreed to the published version of the manuscript.

**Funding:** This research received no external funding.

**Conflicts of Interest:** The authors declare no conflict of interest.

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
