# Peer review of "Sustainable Co-Creation Behavior in a Virtual Community: Antecedents and Moderating Effect of Participant’s Perception of Own Expertise"

_sustainability, doi:10.3390/su12198151_

Round 1

Reviewer 1 Report

I found the article very interesting and useful. The study on new forms of participation (co-creation, co-innovation...) via the web is definitely current and with an even greater value for the future. I believe that from this study we can move on to similar analyzes for other sectors, such as online newspapers (we know of the crisis of world journalism) or, if desired, also local administrations on specific projects. In this regard, I remember the case of Tuscaloosa, a small town in Alabama, which in 2011 was partially destroyed by a tornado, but then rebuilt with a good participation of the residents, who participated in the online discussion on the new urban plan of the city. In short, the study seems to me very good and with prospects for further development.

Author Response

Thank you very much for your comments. 

Reviewer 2 Report

Dear Authors,

I found your work interesting. Value co-creation is a much debated topic in marketing literature and scholars are increasingly recommending more and more empirical studies for its operationalization. The analysis of value co-creation in virtual communities makes the research original and interesting.

The literature review is updated and exhaustive; the sections for the construction of the research hypotheses and the methodology adopted are appreciable. The results have been well discussed and interesting practical implications have been derived.

I would have only a few minor suggestions that could improve the work further: it is better to write co-creation instead of cocreation; in the literature section co-creation behaviors could be better described by mentioning the different categories of Yi and Gong (2013), which propose the model for the analysis of co-creation behaviors more used by the scholars and also by you to analyze the advocacy behavior; in the last section you could better specify the implications of the results in general, and specifically for brand management.

Author Response

Thank you very much for your comments. First, we write co-creation instead of cocreation through all the text. Second, we have improved the literature section about co-creation behaviors incorporating Yi & Gong (2013) study (see page 3, lines 104-109 in red). Also, we specify better the implications of the results for brand management (see last paragraph page 15).

Reviewer 3 Report

This is a sound contribution that addresses a relevant research topic,  covers latest literature and employes a solid research method and appropriate data analysis. 

I have some minor suggestions :

  • Change 11 point Likely scale to 10 point Likert scale in method section, this seems as a thypo. 
  • Justify the use of AMOS. 
  • Report the HTMT values for discriminant validity.
  • Proof read the paper, some statements need adjustments (as the final sentence on page 7, for example).  

Author Response

Thank you very much for your comments. We have changed the thypo about the scale (lines 362-363). Also, we have justified the use of AMOS (lines 369-372 in methodology section). We have made adjustments in some statements (for example, final sentence on page 7). We have incorporated HTMT values in Table 3 (values in red above the diagonal).